# Morphological changes in amblyopic eyes in choriocapillaris and Sattler's layer in comparison to healthy eyes, and in retinal nerve fiber layer in comparison to fellow eyes through quantification of mean reflectivity: A pilot study

Oussama Samer Masri[1], Bachir Abiad[2], Mouhamad J. Darwich[3], Paulette Antonios Sarkis[2], Georges M. El Mollayess[2], Zeina Nasser[1], Youssef Fares[1], Elie Al Ahmar[4,5], Elias Estephan[1,4] *

**1** Faculty of Medical Sciences, Neuroscience Research Center, Lebanese University, Beirut, Lebanon, **2** Ophthalmology Department, Monla Hospital, Tripoli, Lebanon, **3** Faculty of Sciences, Lebanese University, Fanar, Lebanon, **4** Faculty of Arts and Sciences, Holy Spirit University of Kaslik, Kaslik, Lebanon, **5** School of Engineering, Holy Spirit University of Kaslik, Kaslik, Lebanon

* eliasestephan@usek.edu.lb

## Abstract

### Purpose

Establishing the reliability of a new method to check the mean retinal and choroidal reflectivity and using it to find retinal and choroid changes in amblyopia.

### Methods

Design: Retrospective case-control. Population: 28 subjects of which 10 were healthy controls (20 eyes): 8 with refractive errors, 1 with strabismus, and 1 with both. 18 patients with unilateral amblyopia included: 7 anisometropic, 6 isoametropic, 1 strabismic, and 4 combined. Mean participants' age: 13.77 years ± 10.28. Observation procedures: SD-OCT and ImageJ. Main outcome measure: mean reflectivity of retinal and choroid layers. Amblyopic, fellow, and healthy eyes were compared.

### Results

The method of measuring reflectivity is good to excellent reliability for all regions of interest except the fourth. The mean reflectivity of the choriocapillaris and Sattler's layer in amblyopic eyes were significantly lower than in healthy eyes (p = 0.003 and p = 0.008 respectively). The RNFL reflectivity was lower than that of fellow eyes (p = 0.025). Post-hoc pairwise comparisons showed statistically significant differences between amblyopic and healthy eyes for choriocapillaris (p = 0.018) and Sattler's (p = 0.035), and between amblyopic and fellow eyes for RNFL (p = 0.039).

**Data Availability Statement:** All data files are available from the figshare database (https://doi.org/10.6084/m9.figshare.14906772).

**Funding:** The authors received no specific funding for this work.

**Competing interests:** The authors have declared that no competing interests exist.

## Conclusion

A decrease in reflectivity of the choriocapillaris and Sattler's in amblyopic compared to healthy eyes, and a decrease in reflectivity of the RNFL in the amblyopic compared to fellow eyes, indicate that the pathophysiology is partly peripheral and might be bilateral.

## Introduction

Amblyopia is one of the most important causes of pediatric vision impairment. The estimated prevalence of amblyopia is 2–4% in the general population [1]. Age, height, gender, ethnicity, and socioeconomic status do not affect the occurrence of amblyopia [2].

The neuropathophysiology behind the development of amblyopia is not yet fully understood [1]. Although defined as a disease with no organic findings [1], many studies have detected changes affecting the retina [3–8], lateral geniculate nucleus [9, 10], and the visual cortex [9, 11]. Whether the pathological mechanism originates in the retina, primary cortex, or higher centers remains a matter of controversy [11], as well as whether amblyopia is a monocular or binocular condition [1, 12].

Multiple factors can induce amblyopia: refractive errors, misalignment, opacities, or occlusions.

This research aims to determine any significant structural differences found in the amblyopic retina and choroid, in comparison to fellow eyes (FE) and healthy eyes (HE), through mean reflectivity quantification. This will help us determine whether the condition is unilateral or bilateral and whether it is peripheral (ie in the retina and choroid) or central (intracranial visual pathway, visual centers, and higher visual networks). To our knowledge, this is the first OCT study that has mean reflectivity as the main outcome measure.

## Materials and methods

This is a case-control study performed at Monla Hospital eye center, Tripoli, Lebanon from March 2018 to October 2019. All participants were treated in accordance with the tenets of the Declaration of Helsinki. Verbal and written informed consent were obtained from all participants or parents of minors. This study was carried out after approval from the Institutional Board of the Lebanese University. We examined 39 patients. Patients with organic brain disease, neurological disorder, or organic eye disease (history of trauma, intraocular surgery, laser treatment, glaucoma or retinal disorders), deprivation amblyopia, bilateral amblyopia, successful treatment for previously diagnosed amblyopia, or uncooperative for examination were excluded. Accordingly, 21 were excluded, and the final sample size was 18 patients with unilateral amblyopia. All were diagnosed at the time of image acquisition. We examined 20 healthy eyes from controls matched for age and type of ocular abnormality (refractive error and strabismus).

A comprehensive eye exam was performed, including BCVA using a Snellen chart at 6m, slit-lamp examination, automated refraction, manifest refraction, cycloplegic retinoscopy and dilated fundus examination, 10-D fixation test, Hirschberg test, extraocular motility assessment, cover and cover-uncover tests, and intraocular pressure assessment. Amblyopia was defined as a BCVA of 20/30 or worse and/ or at least a two-line difference between the two eyes' BCVA. In this study, anisometropia was defined as an interocular difference of ($>$2D) of myopia, ($>$1D) of hyperopia, or ($>$1D) of astigmatism. For Isoametropia, criteria included hyperopia of ($>$+2.50D) and astigmatism of ($>$+1.75D).

All OCT images were obtained using SD-OCT (Avanti RTVue XR, 2018.0; Optovue). A 12 x 4 mm rectangle covering the macula and optic nerve head (ONH) was chosen to take 12 raster line scans, of which the central line which is passing through the fovea was selected to do the calculations. The eyes were scanned in dim light. The right eye was scanned first followed by the left. Internal fixation was used to ensure proper eye alignment. In case of poor fixation, the image was centered on the macula according to the fundus image generated by the machine. Tracking mode and Motion Correction Technology were employed to eliminate blinking and eye motion artifacts. For proper tracking, we ensured good quality infra-red images. All imaging was performed by a blinded experienced photographer. The selected slice was saved in greyscale, JPEG format.

We elaborated a standardized process of measurement. All images were analyzed using ImageJ 1.52n. After launching the software, we opened the OCT images by dragging and dropping them into the imageJ program, then we set the scale in pixels/microns using the scale bar present on the OCT slices. To do so, we picked the "straight line" from the ImageJ toolbar, and then drew a line along the scale bar present on the OCT image S1 Fig in S1 File. To make sure that the measurement is precise we zoomed in while drawing the line. The software provides the angle of the line which should be set at 90 degrees. The angle can be maintained by holding down the shift button. The software will also determine the length of any drawn line automatically. We selected "analyze" and then "set scale", we entered the number written on the scale bar (in our case 250) in the box next to "Known distance", and we entered "microns" in the box next to "Unit of length" and checked the box "Global", finally we pressed "Ok" S2 Fig in S1 File. The external limiting membrane and ellipsoid zone peaks were used as hallmarks to define the center of the retina. we drew a vertical line through these structures S3 Fig in S1 File, and 2 more parallel lines were drawn 845 microns away from it, one to its right and another to its left S4 and S5 Figs in S1 File then, we made the lines permanent by pressing "edit" then "draw" right after drawing each of them. We used these two lines as borders within which, we delineated the first 6 regions of interest (ROI). We clicked on "analyze", "tools" and opened the "ROI manager". Using the "polygon selections" on the main toolbar, we delineated the ROIs (mentioned below). We added each ROI to the ROI manager by clicking "add" on the "ROI manager" menu right after delineation. We measured the area, mean standard deviation, minimal and maximal pixel intensity of each ROI by selecting all added ROIs and clicking "measure".

ROIs details: ROI#1: Henle fiber layer and ONL, external limiting membrane, myoid zone, ellipsoid zone and outer segments of photoreceptors S6 Fig in S1 File, ROI#2: from Henle fiber layer up to (including) the myoid zone S7 Fig in S1 File, ROI#3: the interdigitation zone plus the retinal pigment epithelium (RPE)/ Bruch's membrane S8 Fig in S1 File, ROI#4: choriocapillaris and Sattler's layer S9 Fig in S1 File, ROI#5: the choriocapillaris S10 Fig in S1 File, ROI#6: vitreous (rectangle: 1690 microns in width and 200 microns in height) S11 Fig in S1 File. The latter was needed for normalization as noise and signal to noise ratio will change from acquisition to acquisition. The vitreous was selected for this purpose since, in this region, the OCT signal is weak (no backscatterers) and therefore the noise is most prominent. For the remaining ROIs, the first border was a line 90 microns away from the temporal edge of the ONH, it was drawn perpendicular to the x-axis of the image. A line parallel to this one was drawn 960 microns away from the ONH edge S12 Fig in S1 File. Within these borders, the following ROIs were chosen: ROI#7: the nasal ganglion cell complex S13 Fig in S1 File, ROI#8: the retinal nerve fiber layer (RNFL) S14 Fig in S1 File, ROI#9: the ganglion cell layer (GCL) and inner plexiform layer (IPL) S15 Fig in S1 File, ROI#10: the IPL and the inner nuclear layer S16 Fig in S1 File, and ROI#11: the outer plexiform layer S17 Fig in S1 File. We calculated the mean reflectivity in each region using ImageJ, normalized with ROI#6, and later compared

these ratios using SPSS. In addition, we calculated and compared the area of ROI#12 S18 Fig in S1 File; the upper border was a straight line connecting the two macular peaks to each other, and the lower border was a segmented line drawn along the vitreoretinal interface starting from one peak and ending at the other. All of the ROIs were drawn manually. The IN-OCT consensus was referenced for nomenclature [13]. The measurer of ROIs was blinded. (All S1 File, where x represents the figure's number as well as all original OCT images and their Ima-geJ analysis are provided as S1 File)

### Analysis of repeatability

To check for inter-rater reliability, four independent raters followed the instructions of the standardized process of measurement to measure 12 ROIs each from 10 randomly selected participants. To check for intra-rater reliability, one rater repeated the measurements four times across all 12 ROIs in a randomly selected participant. The interpretation of the reliability follows the guidelines (ICC < 0.5: poor; 0.5 < ICC < 0.75: moderate; 0.75 < ICC < 0.9: good; 0.90 < ICC: excellent reliability) [14].

### Statistical analysis

Statistical analyses were performed using the SPSS v25. To assess the intra and inter-rater reliability, we calculated two intraclass correlation coefficients (ICC) based on the McGraw and Wong (1996) Convention [15]. The ICC for inter-rater reliability calculated based on single rating, absolute agreement, 2-way mixed effects model. The ICC for inter-rater reliability was calculated based on a single rating, absolute agreement, 2-way mixed-effects model [14].

In the descriptive statistics, continuous variables were expressed in means and standard deviations, while categorical variables were expressed in proportions and frequencies. The area of ROI#12 and reflectivity ratio of remaining ROIs of the 3 groups of interest: amblyopic eye (AE), FE, and HE were calculated. The independent sample t-test was used to compare 2 groups when data were approximately normally distributed and had no outliers. When those assumptions were violated, the Mann-Whitney test was used. When comparing the ROIs of the 3 groups, a one-way ANOVA was used when the conditions of normality and homogeneity of variances were not violated, otherwise, we used Kruskal-Wallis. Effect sizes were calculated using formulae from this paper [16]. These included Cohen's d with the Hedge's g correction, r, $r^2$ or $\eta^2$, and $\varepsilon^2$. Post-Hoc Pairwise Comparisons were conducted for the statistically significant results of the Kruskal-Wallis H test, and these significant values were adjusted by the Dunn-Bonferroni correction for multiple tests. We assessed normality using the Shapiro-Wilk test, and homogeneity of variances with Levene's test. We used boxplots to check for outliers. The asymptotic significance (2-sided tests) was set at a p-value of <0.05.

### Results

Delineating the ROIs is done in a freehand manner which is a source of error in measurement.

The raters found the method to be simple and didn't require much training. The results of measurements of various raters is shown in Table 1.

For ROIs with a CoV less than 3% (all expect ROI#4), The ICC estimate for inter-rater reliability ranged from 0.912 for ROI#5 (95% CI [0.795–0.974], CoV = 2.7%) up to 0.96 (ICC for ROI#3 = 0.956, 95% CI [0.876–0.988], CoV = 0.42%). For ROI#4, the ICC estimate was 0.375 (95% CI [0.092–0.726]). This indicates this method has good to excellent reliability for all ROIs, except for ROI#4. The ICC estimate for intra-rater reliability for the first rater was 0.994, and its 95% confidence interval was [0.986; 0.998].

**Table 1. Mean results, standard deviation and coefficient of variation of the reflectivity of ROIs#1–11 and area of ROI#12 of different raters.**

| Region of interest | Rater 1 | Rater 2 | Rater 3 | Rater 4 | All Raters | |
|---|---|---|---|---|---|---|
| | Mean ± SD | | | | Mean ± SD | CoV |
| ROI#1 | 94.39 ± 44.29 | 98.00 ± 46.05 | 96.05 ± 44.14 | 98.36 ± 45.85 | 96.70 ± 1.84 | 1.90 |
| ROI#2 | 75.05 ± 18.47 | 76.35 ± 19.30 | 77.90 ± 20.46 | 78.84 ± 21.82 | 77.04 ± 1.67 | 2.17 |
| ROI#3 | 191.65 ± 26.25 | 193.23 ± 26.13 | 193.02 ± 26.20 | 191.84 ± 26.54 | 192.43 ± 0.81 | 0.42 |
| ROI#4 | 98.92 ± 23.83 | 95.68 ± 27.60 | 87.96 ± 28.40 | 76.51 ± 30.99 | 89.77 ± 9.96 | 11.10 |
| ROI#5 | 125.12 ± 15.32 | 133.26 ± 17.67 | 127.18 ± 15.87 | 130.00 ± 14.30 | 128.89 ± 3.54 | 2.74 |
| ROI#6 | 12.91 ± 7.42 | 13.13 ± 7.49 | 13.12 ± 7.50 | 13.14 ± 9.26 | 13.07 ± 0.11 | 0.85 |
| ROI#7 | 133.89 ± 33.72 | 133.76 ± 33.53 | 131.54 ± 33.36 | 133.08 ± 33.69 | 133.07 ± 1.08 | 0.81 |
| ROI#8 | 167.66 ± 21.55 | 166.65 ± 21.13 | 165.86 ± 22.87 | 169.02 ± 19.97 | 167.30 ± 1.36 | 0.82 |
| ROI#9 | 108.82 ± 15.76 | 110.05 ± 16.63 | 109.89 ± 16.48 | 111.12 ± 17.98 | 109.97 ± 0.94 | 0.86 |
| ROI#10 | 102.50 ± 19.29 | 102.39 ± 19.27 | 103.12 ± 18.72 | 102.91 ± 18.85 | 102.73 ± 0.34 | 0.34 |
| ROI#11 | 111.67 ± 14.43 | 110.50 ± 14.71 | 112.40 ± 14.38 | 110.26 ± 15.22 | 111.21 ± 1.00 | 0.90 |
| | Area | | | | Mean area ± SD | CoV |
| ROI#12 | 41540.80 | 40881.08 | 41862.08 | 42504.34 | 41697.07 ± 675.57 | 1.62 |

SD: standard deviation.

CoV: coefficient of variation (in percentage).

A total of 18 subjects met the inclusion criteria and were compared to 10 controls. Out of the 28 participants, 20 (71.43%) were Lebanese and 8 (28.57%) were Syrian. The mean age was 14.61 years ± 11.41 ranging from 3 to 55 years. Among them, 17 (60.71%) were males and 11 (39.29%) were females. 21 participants had only refractive errors (75%), 2 (7.14%) had only strabismus, and 5 had both (17.86%). Among participants we had 14 hyperopes (5 healthy and 9 amblyopic) and 7 myopes (3 healthy and 4 amblyopic). Table 2 shows the socio-demographic and clinical characteristics of the participants.

We compared all the ROI ratios and ROI#12 area described earlier, of AE to the FE, then AE to HE, and finally FE to HE. The statistically significant results of these tests are displayed in Table 2. When comparing FE to HE, we found no statistically significant differences. However, when comparing AE to FE, we found significant differences only in ROI#7 and ROI#8 ($p = 0.034$, $\eta^2 = 0.125$, and $p = 0.025$, $\eta^2 = 0.14$ respectively). Both ROIs are less reflective in AE than in FE as Table 3 shows.

When comparing AE to HE, we found that the mean reflectivity of AE (6.54 ± 1.06) was significantly lower than the mean reflectivity of HE (7.57 ± 0.91), $t(36) = 3.201$, $p = 0.003$, 95% CI

**Table 2. Demographics and clinical characteristics of participants.**

| Sample size | Total (N = 28) | | Patients (n = 18) | Healthy (n = 10) |
|---|---|---|---|---|
| Age | Range (3 to 55) | | 3 to 55 | 4 to 28 |
| | Mean ± SD (14.61 ± 11.41) | | 15.89 ± 13.41 | 12.30 ± 6.48 |
| Sex | Male | 17 (60.71%) | 9 (50%) | 8 (80%) |
| | Female | 11 (39.29%) | 9 (50%) | 2 (20%) |
| Nationality | Lebanese | 20 (71.43%) | 11 (61.11%) | 9 (90%) |
| | Syrian | 8 (28.57%) | 7 (38.89%) | 1 (10%) |
| Type of eye disease | Anisometropia | 21 (75%) | 7 (38.89%) | 8 (80%) |
| | Isometropia | | 6 (33.33%) | |
| | Strabismus | 2 (7.14%) | 1 (5.56%) | 1 (10%) |
| | Combined | 5 (17.86%) | 4 (22.22%) | 1 (10%) |

**Table 3. Comparison of the reflectivity of ROIs 4,5,7 and 8 among AE, FE, and HE.**

| ROI | Amblyopic Eye (n = 18) | | | Fellow Eye (n = 18) | | | Healthy Eye (n = 20) | | | AE-FE | AE-HE | FE-HE |
|---|---|---|---|---|---|---|---|---|---|---|---|---|
| **4** | 6.54 | ± | 1.06 | 7.07 | ± | 1.08 | 7.57 | ± | 0.91 | | **0.003*** | |
| | 6.77 (1.47) | | | 7.33 (1.61) | | | 7.52 (1.55) | | | 0.137 | | 0.306 |
| **5** | 8.88 | ± | 1.57 | 9.65 | ± | 1.58 | 10.09 | ± | 1 | | | 0.316 |
| | 9.07 (1.69) | | | 10.20 (2.47) | | | 10.15 (1.44) | | | 0.107 | **0.008*** | |
| **7** | 8.84 | ± | 2.17 | 10.09 | ± | 2.11 | 10.22 | ± | 1.41 | | | |
| | 8.91 (2.44) | | | 11.01 (2.89) | | | 10.25 (2.54) | | | **0.034*** | **0.028*** | 0.838 |
| **8** | 11.31 | ± | 2.61 | 12.95 | ± | 2.46 | 12.82 | ± | 1.58 | | | |
| | 11.72 (2.80) | | | 13.84 (3.27) | | | 13.06 (2.63) | | | **0.025*** | **0.044*** | 0.563 |

The two-by-two comparisons are amblyopic to fellow eye (AE-FE), amblyopic to healthy (AE-HE), and fellow to healthy (FE-HE).

Values are: mean ± SD in the upper row, and median (IQR) in the lower row of each ROI. IQR is the interquartile range.

When the p-value is displayed in an upper row, it is the p-value of an independent sample t-test and when displayed in a lower row, it is the p-value of the non-parametric Mann-Whitney test.

*p-value < 0.05 is considered significant.

[0.375; 1.674], with a large effect size (Hedge's g = 1.04, $r^2$ = 22.16%): the mean reflectivity of ROI#4 in AE is 1.04 SDs below the corresponding reflectivity in HE. We also found that the reflectivity of ROIs#5,7 and 8 was statistically significantly lower in AE in comparison to HE (U = 89, p = 0.008, r = 0.431, $\eta^2$ = 0.186; U = 105, p = 0.028, r = 0.356, $\eta^2$ = 0.127; U = 111, p = 0.044, r = 0.327, $\eta^2$ = 0.107).

When comparing the 3 groups to each other, there were statistically significant differences in reflectivity of ROI#4 ($\chi^2$(2) = 7.636, p = 0.022, $\varepsilon^2$ = 0.139), ROI#5 ($\chi^2$(2) = 6.672, p = 0.036, $\varepsilon^2$ = 0.121), and ROI#8 ($\chi^2$(2) = 6.567, p = 0.038, $\varepsilon^2$ = 0.119) (Table 4).

After conducting post-hoc pairwise comparisons, we found a statistically significant difference between AE and HE for ROI#4 (Adjusted p = 0.018) and ROI#5 (Adjusted p = 0.035). With the reflectivity of these layers being lower in AE than in HE. We also found that the reflectivity of ROI#8 in AE is statistically significantly lower than its counterpart in FE (Adjusted p = 0.039). Even though we found statistical significant difference in ROI#7 in the multiple comparison step (p = 0.045), we found no significan differences between any two groups after adjustement of the p-value in the post-hoc analysis.

To account for any possible influence of axial length on the measurements we subdivided the sample into two subgroups: hyperopes and myopes. The hyperopic group included 38 eyes (12 HE, 13 AE and 13 FE). The myopic group included 14 eyes (6 HE, 4 AE and 4 FE) and was therefore too small for analysis.

In hyperopic participants, when comparing the AE to the HE, we found that the reflectivity of ROI#4 was statistically significantly lower in the AE than the HE (U = 38, p = 0.03, r = -0.435,T $\eta^2$ = 0.1893). When comparing the AE to the FE, we found that the reflectivity of

**Table 4. Comparison of the reflectivity of the ROIs#4, 5, and 8 among amblyopic, fellow, and healthy eyes.**

| | Amblyopic | Fellow | Healthy | Kruskal-Wallis H | p-value | $\eta^2$ (%) | $E_R^2$ |
|---|---|---|---|---|---|---|---|
| **ROI#4** | 6.77 (1.47) | 7.33 (1.61) | 7.52 (1.55) | 7.636 | **0.022*** | 10.63 | 0.139 |
| **ROI#5** | 9.07 (1.69) | 10.20 (2.47) | 10.15 (1.44) | 6.672 | **0.036*** | 8.82 | 0.121 |
| **ROI#8** | 11.72 (2.80) | 13.84 (3.27) | 13.06 (2.63) | 6.567 | **0.038*** | 8.62 | 0.119 |

Values are median (IQR). IQR is the interquartile range

*p-value < 0.05 is considered significant.

**Table 5. Comparison of the reflectivity of the ROIs#4, 5, 7 and 8 among amblyopic, fellow, and healthy eyes in hyperopic patients.**

|  |  | Hyperopic | | | p-value |
|---|---|---|---|---|---|
|  |  | AE | FE | HE |  |
|  | n | 13 | 13 | 12 |  |
| ROI#4 | Mean ± SD | 6.81 ± 0.89 | 7.35 ± 0.83 | 7.78 ± 1.04 |  |
|  | Median (IQR) | 7.14 (0.96) | 7.56 (1.10) | 7.90 (1.88) | 0.077 |
| ROI#5 | Mean ± SD | 9.28 ± 1.23 | 10.04 ± 0.98 | 10.09 ± 1.18 |  |
|  | Median (IQR) | 9.50 (1.57) | 10.28 (1.35) | 10.22 (1.97) | 0.192 |
| ROI#7 | Mean ± SD | 9.51 ± 1.51 | 10.84 ± 1.19 | 9.99 ± 1.40 |  |
|  | Median (IQR) | 9.55 (2.68) | 11.36 (1.17) | 10.25 (2.29) | **0.039***|
| ROI#8 | Mean ± SD | 12.06 ± 1.68 | 13.79 ± 1.42 | 12.57 ± 1.83 |  |
|  | Median (IQR) | 11.94 (3.04) | 14.14 (1.71) | 12.81 (3.03) | **0.031***|

*p-value < 0.05 is considered significant.

ROI#7 and ROI#8 were statistically significantly lower in the AE than the FE (U = 37, p = 0.015, r = -0.477, $\eta^2$ = 0.2282; and U = 33, p = 0.008, r = -0.517, $\eta^2$ = 0.2683 respectively).

When comparing the 3 groups to each other, there were statistically significant differences in reflectivity of ROI#7 ($\chi^2$(2) = 6.502, p = 0.039, $\eta^2$ = 0.1286, $\varepsilon^2$ = 0.1757), and ROI#8 ($\chi^2$(2) = 6.923, p = 0.031, $\eta^2$ = 0.1406, $\varepsilon^2$ = 0.1871) (Table 5).

After conducting post-hoc pairwise comparisons, we found a statistically significant difference between AE and FE in ROI#7 (adj p = 0.037), and in ROI#8 (adj p = 0.03), with the reflectivity being lower in the AE than in the FE in both ROIs.

## Discussion

We present a new method of assessing the retina and choroid by measuring the reflectivity means of their layers using OCT. We elaborated a standardized process of measurement, which allowed reproducibility. Its repeatability, intra- and inter-rater variability are yet to be determined conclusively as per the guideline by Koo and Li [14]. Our results indicate, that the method's reliability ranges from good to excellent for all ROIs, but it has poor reliability for assessing Sattler's layer where the highest variability between raters was found (Coefficient of Variation = 11.1%). The second highest coefficient of variation was only 2.74%. For these regions, the ICC was higher than 0.9. If the steps of the standardized process method are followed, the resulting variations between raters are acceptable. We attribute the higher variability in measuring Sattler's to the lack of clear demarcation between Sattler's and Haller's layer [13].

A future study dedicated to specifically investigating the reliability of this technique should be performed.

Many studies tried to uncover organic changes underlying amblyopia by inspecting different structures: from the retina up to the primary visual cortex and higher visual networks [1]. We focused on the retina and underlying choroid choosing SD-OCT as a neuroimaging technique. Previous OCT studies mostly focused on thickness. To our knowledge, this is the first study that uses mean reflectivity as an outcome measure. We found the mean reflectivity of the choriocapillaris (ROI#4) and Sattler's layer (ROI#5) significantly lower in AE than in HE, and that of the RNFL (ROI#8) lower in AE than that of FE.

Many factors affect reflectivity: Hard exudates, calcifications, hemorrhages, vitreous membranes, choroidal neovascularization, inflammatory infiltrates, scars and fibrosis [17], melanin [18], mitochondria [13] increase reflectivity. Decreased cellular density, fluid accumulation,

signal attenuation, cellular orientation decrease reflectivity. Cellular orientation may lead to either.

As for the choroid, an ultrasound study showed that the resistive index of posterior ciliary arteries is not different between AE and FE [19]. This reflects a normal resistance to blood flow within the choroid. A study using OCT demonstrated that the subfoveal choroidal thickness (CT) is statistically the same in both AE and FE [20]. But contrary to these findings, other OCT studies have often shown that the subfoveal CT is increased in AE [21]. It is difficult to compare other OCT studies to ours since most of them studied thickness which, in many cases, does not affect the mean reflectivity of a given structure. However, in the case of the choroid, we can say that when it becomes thicker, deeper layers will suffer from more signal attenuation as the distance away from the RPE increases [17]. Hence, we expected the deeper pixels of Sattler's layer (ROI#4) to be less reflective in AE in comparison to both FE and HE (with no difference between FE and HE). Fig 1 illustrates this concept. We expected a similar unilateral decrease in the reflectivity of the choriocapillaris (ROI#5) due to atrophy [22].

In our study, we detected changes only when comparing AE to HE in both ROI#4 and 5: the reflectivity of ROI#4 was lower in AE compared to HE ($p = 0.003$), but not when compared to FE ($p = 0.137$). The difference between them ranged from 0.375 to 1.674 on average, with a large effect size (Hedge's $g = 1.04$, $r^2 = 22.16\%$), with 22.16% of the variability in reflectivity explained by amblyopia. There was a similar difference in ROI#5 between AE and HE ($p = 0.008$) with a more moderate effect size ($r = 0.431$, $\eta^2 = 18.6\%$). These findings were also detected in the post-hoc analysis of the Kruskal-Wallis test ($p = 0.018$ and $p = 0.035$ for ROI#4 and 5 respectively). This is evidence supporting the hypothesis that amblyopia may not be merely a functional disease, but may have an organic component in the choriocapillaris and Sattler's layer. This might be clinically useful for early screening and detection of amblyopia, giving this technique potential diagnostic value (i.e. if the reflectivity of ROI#4 is 1 SD below the normal mean, this might be indicative of amblyopia), as well as a tool to follow-up the effectiveness of an intervention or treatment for amblyopia. This needs other studies dedicated to determining the sensitivity and specificity of this tool. Since no difference was detected between AE and FE, this implies that FE does not equal HE, leading us to believe that the changes cannot be unilateral. A possible explanation is a bilateral decrease that is affecting AE

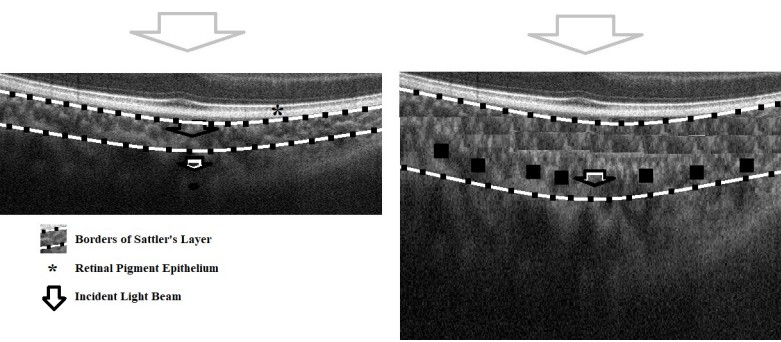

**Fig 1. Correlation between choroidal thickness and reflectivity.** This diagram shows a HE on the left and an AE on the right with exaggerated increased thickness. The arrows in the left image show how the incident OCT beam gradually decreases as it moves beyond the RPE. Highly reflective structures such as RPE can lead to attenuation of an incident signal. Therefore, the choroid that falls posterior to the RPE, is relatively hyporeflective [17]. Assuming that both eyes have the same signal-to-noise ratio, the arrow with the darker borders shows the same level of signal attenuation in both eyes. In the HE, the attenuation did not affect Sattler's layer as much as it affected it in the AE where the layer is thicker (more black pixels represented by black boxes). Haller's layer would suffer from even more attenuation in comparison to Sattler's.

more than FE. The change in FE was not large enough to be detected by our statistical analysis (we observed a gradual decrease of reflectivity going from HE to FE to AE Table 2).

As for the retina, previous histologic studies in amblyopic animal models have demonstrated atrophy in optic nerve fibers and a decrease in retinal ganglion cells' density and size [19, 23, 24], in addition to a decrease in their nucleolar volume [25]. The amacrine synapses found within the IPL of amblyopic eyes were increased [26]. In light of histological [10, 19, 23–25] and OCT studies that supported the presence of retinal morphological abnormalities [3–8], we expected to find significant changes in the mean reflectivity of AE. We hypothesized to see a decrease in the RNFL (ROI#8) and GCL (ROI#7 and 9), and a change in the average intensity of IPL (ROI#7 and 10). We only found significant changes in ROI#8. It was less reflective in AE in comparison to FE (p = 0.025) with a medium effect size (r = 0.374, $\eta^2$ = 14%). These findings were also detected in the post-hoc analysis of the Kruskal-Wallis test (p = 0.039). Contrary to our expectations, there were no statistically significant changes in mean reflectivity scores of ROIs#7, 9 and 10.

Since no differences were detected between the reflectivity scores of ROI#8 in AE and HE, FE and HE cannot be equal, leading us to believe that the condition is more likely to be bilateral. A possible explanation is that amblyopia is affecting both eyes in different ways: the cellular density in RNFL is decreased in AE and it is increased in FE as a compensatory mechanism [27].

The above may be explained, as well, by a change in cellular orientation in each layer since that affects reflectivity. Certain layers such as the RNFL, Henle's fiber layer, and the photoreceptor layer have angle-dependent optical reflectivity properties [28]. Some eye diseases—such as glaucoma, age-related macular degeneration, and macular edema—may affect the directional reflectivity properties of some layers.

These findings can help us shed light on whether amblyopia is unilateral or bilateral in nature. We explain our expectations and results in Fig 2.

The supraretinal area (ROI#12) that we selected lies near the surface of the retina, and changes affecting this area would have reflected morphological changes in the foveal pit and surrounding peaks we found no significant changes.

These findings do not align both with the histological [10] and OCT neuroimaging studies [20] that haven't found any ocular morphological change, and with the opinion that amblyopia is purely a central condition that affects the visual cortex and higher visual networks while sparing the retina and the choroid [1, 12, 20]. When it comes to choroidal and retinal changes, our study showed that these changes might be bilateral in nature which aligns with literature suggesting the bilateral nature of the condition [1, 10, 12].

To account for any potential effect of the axial length, we repeated the analyses on a subset of participants that excluded myopic and strabissmic emmetropic participants, leaving hyperopes. We found a statistically significant difference in the RNFL with the mean reflectivity of the FE significantly higher than the mean reflectivity of the AE which is consistent with the previous finding. Similar consistency was found in the difference between the AE and the HE in the choroid (namely ROI#4: choriocapillaris and Sattler's) with the reflectivity of the AE being statistically significantly lower than that of the HE using the Mann-Whitney test. However this was not found using the Kruskal Wallis test, this might be due to the smaller sample size and/or the difficulty in delineating the borders of ROI#4. It could also be attributable to differences brought about by different axial lengthes that are associated with changes in choroidal thickness [29] and volume [30], which in turn could be affecting the reflectivity.

Despite conducting the study in only one center in a relatively low-income area, the final sample size is decent. The technology used is optimal, as SD-OCT is superior to TD and SS-OCT when it comes to studying the macula (especially if SS-OCT's wide field is not needed)

**Fig 2. Juxtaposition of hypothesized results vs actual findings.** Our offered hypotheses for possible changes in amblyopia are shown on the left: If the change in a layer is unilateral, we should find significant differences between AE and FE, as well as between AE and HE, since FE would be unaffected. If the condition is bilateral and affects both eyes equally: we should find differences either between AE and HE or between FE and HE. If both eyes are affected but unequally, we should find differences in all comparisons, and in which case, the AE is likely to be more affected than FE. On the right, we illustrate our findings: in the case of the choriocapillaris and Sattler's, their reflectivity was lower only in AE when compared to HE, while in the case of RNFL, the difference was between AE and FE only.

[31]. The technique we used to assess for morphological changes was never used before in literature. To our knowledge, this technique is used for the first time in Lebanon and worldwide, and we report effect sizes for any found significant differences.

Our findings should be considered within the confines of several limitations: the sample size is small and didn't allow to adjust for potentially confounding variables such as sex and age, and there were neither age nor refractive error range restrictions. The study was not limited to one type of amblyopia, one type of strabismus, nor one type of refractive error. We did not exclude those with high diopters because of the small sample size. We don't have normative values for mean reflectivity nor do we know of any diurnal variations that may affect it. Diurnal changes, for example, affect the choroid thickness, and if not taken into consideration can result in inaccurate conclusions [20]. We accounted for this by taking the measurements in the same time period. For visual acuity, the Snellen chart was used which is a non-logMAR scale less accurate than its logMAR counterpart [32]. The chosen OCT sections were saved as JPEG images which leads to a loss of quality, subtle changes in reflectivity intensity, and its location [33]. The best we could do regarding this issue is to avoid any further manipulations or compressions before measuring our ROIs.

## Conclusion

In this study, we sought to detect morphological changes in the eyes of patients with unilateral amblyopia (anisometropia, isometropia, strabismus and combined). Using SD-OCT, we detected a decrease in the reflectivity of the choriocapillaris and Sattler's layer in AE when compared to HE, and a decrease in reflectivity of the RNFL in AE when compared to the FE. Our results indicate that the pathophysiology has a peripheral element that might be bilateral in nature, though more research is warranted. Future research should be conducted using larger sample sizes with stricter exclusion criteria and more homogeneous groups that adjust for age, nationality, refractive error and type of amblyopia. Other types of amblyopia should be taken into account, such as deprivation amblyopia. Also, future research should use the newer SS-OCT technology instead of the SD-OCT when studying the choroid since it is better suited for that purpose (no signal attenuation beyond the RPE). This research serves as a platform to encourage the development of integrated software for OCTs that may facilitate this kind of measurements in the clinical and academic settings. Clinically, this tool may be used to objectively detect amblyopia. Furthermore, it can be used to research other pathologies. Additional research is needed to determine a dose-response relationship, consequently these measurements may be used for follow-up.

## Supporting information

**S1 File. Measuring method and RIOs.**
(DOCX)

## Acknowledgments

Rudy Chidiak, Abdulrahman El Sayed and Inas Masri.

## Author Contributions

**Conceptualization:** Bachir Abiad, Youssef Fares, Elie Al Ahmar, Elias Estephan.

**Data curation:** Oussama Samer Masri, Paulette Antonios Sarkis, Georges M. El Mollayess.

**Formal analysis:** Oussama Samer Masri, Mouhamad J. Darwich, Zeina Nasser, Elie Al Ahmar, Elias Estephan.

**Investigation:** Oussama Samer Masri.

**Methodology:** Bachir Abiad, Youssef Fares, Elie Al Ahmar, Elias Estephan.

**Supervision:** Elias Estephan.

**Visualization:** Bachir Abiad, Youssef Fares, Elie Al Ahmar, Elias Estephan.

**Writing – original draft:** Oussama Samer Masri.

**Writing – review & editing:** Oussama Samer Masri, Mouhamad J. Darwich, Zeina Nasser, Elie Al Ahmar, Elias Estephan.

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
