## [Decision Letter · Decision Letter 0]

10 May 2021

PONE-D-21-10099

Morphological changes in eyes of amblyopic patients through quantification of mean reflectivity.

PLOS ONE

Dear Dr. Estephan,

Thank you for submitting your manuscript to PLOS ONE. After careful consideration, we feel that it has merit but does not fully meet PLOS ONE’s publication criteria as it currently stands. Therefore, we invite you to submit a revised version of the manuscript that addresses the points raised during the review process.

A learned reviewer has offered several constructive criticism. Manuscript need to be revised appropriately addressing these comments. 

We look forward to receiving your revised manuscript.

Kind regards,

Sanjoy Bhattacharya

Academic Editor

PLOS ONE

Journal Requirements:

Reviewers' comments:

Reviewer's Responses to Questions

**Comments to the Author**

1. Is the manuscript technically sound, and do the data support the conclusions?

Reviewer #1: Partly

2. Has the statistical analysis been performed appropriately and rigorously? 

Reviewer #1: Yes

3. Have the authors made all data underlying the findings in their manuscript fully available?

Reviewer #1: No

4. Is the manuscript presented in an intelligible fashion and written in standard English?

Reviewer #1: Yes

5. Review Comments to the Author

Reviewer #1: In general:

The topic of the article is about the differences of some OCT parameters between amblyopic eyes and healthy eyes which might be of interest. However, as they applied a non-standardised, not widely used reflectivity estimation method on exported OCT images, the article should include detailed description of the reflectivity measurements and calculations and the accuracy and reproducibility of the method used.

Detailed comments and questions:

The title is not clear enough as it is not containing the place of changes in eyes or the reflectivity of which structure.

Abstract

The authors showed significant differences between healthy and amblyopic eyes, however the screening abilities of the detected changes are not examined, so conclusion about it is not supported by the results.

Methods

The definition and measurement/determination method(s) of reflectivity is not described in details in the Methods. However, this would be crucial as the reflectivity is the main outcome measure in this article. Please, provide also the reproducibility and accuracy values for the reflectivity measurements used in this study.

Results

Table 1. There were possibly significant demographic differences between patients and healthy controls (like: maximum age, sex distribution). Please, include statistical comparisons between patients and controls for the parameters and address these differences in the discussion and disclose that these differences do not cause bias.

The interpretation of the results need to be very restrained as the differences were found significant at the level of p<0.05 and only few of them with p<0.01 and none of them at p<0.001.

Discussion

In line 198: “an U/S study” – please, introduce the abbreviation.

The statement in the sentence in lines 205-207 would need supporting literature reference or justification.

There are no figure legends in the manuscript.

For approving a test for screening needs validation, first. Have to know what is the sensitivity, specificity, etc. of the new diagnostic method. And not only a comparison with healthy eyes but comparisons with other eye diseases are necessary to judge before the statement about the usefulness of the new diagnostic modality.

We know that both retinal thickness and choroidal thickness depend on the axial eye length. The longer eyeballs have thinner layers and possibly with different reflectivity. What was the range of the axial eye length of the patients and controls? Were there differences in axial eye length between AE, FE and HE? Need to show if there is a correlation between axial eye length and the reflectivity of the layers.

6. PLOS authors have the option to publish the peer review history of their article (what does this mean?). If published, this will include your full peer review and any attached files.

Reviewer #1: No

---

## [Author Response · Author response to Decision Letter 0]

4 Jul 2021

Reviewer #1: In general:

The topic of the article is about the differences of some OCT parameters between amblyopic eyes and healthy eyes which might be of interest. However, as they applied a non-standardised, not widely used reflectivity estimation method on exported OCT images, the article should include detailed description of the reflectivity measurements and calculations and the accuracy and reproducibility of the method used.

We calculated two ICCs (intraclass correlation coefficients) the first was an intra-rater reliability measure wherein one rater repeated the measurements four times over twelve regions of interest from a randomly chosen participant, the second was an inter-rater reliability measure wherein four independent raters measured twelve regions of interest from a randomly chosen participant. We added this relevant information to the abstract, materials and methods, the results and discussion sections (lines 20-21, 28, 130-141, 158-170, 237-244). All added parts including tables are highlighted in yellow in the revised manuscript.

Detailed comments and questions:

The title is not clear enough as it is not containing the place of changes in eyes or the reflectivity of which structure.

Addressed

Abstract

The authors showed significant differences between healthy and amblyopic eyes, however the screening abilities of the detected changes are not examined, so conclusion about it is not supported by the results.

Addressed. Mention of its screening and diagnostic potential was removed from the abstract.

Methods

The definition and measurement/determination method(s) of reflectivity is not described in details in the Methods. However, this would be crucial as the reflectivity is the main outcome measure in this article. Please, provide also the reproducibility and accuracy values for the reflectivity measurements used in this study.

Addressed. More details were added as well as images in the supporting information section. All added parts are highlighted in yellow in the revised manuscript

Results

Table 1. There were possibly significant demographic differences between patients and healthy controls (like: maximum age, sex distribution). Please, include statistical comparisons between patients and controls for the parameters and address these differences in the discussion and disclose that these differences do not cause bias.

The interpretation of the results need to be very restrained as the differences were found significant at the level of p<0.05 and only few of them with p<0.01 and none of them at p<0.001.

Addressed. We are in total agreement, and we clarified in the discussion section that we were not able to adjust for these possible confounding factors due to the small sample size. Furthermore, the p-value is affected by the sample size: the larger the sample size, the smaller the p-value will be. We sought to address this by reporting the effect sizes of significant findings.(lines 149, 187, 281, 284, 305, 352) Unlike significance tests, effect size is independent of sample size.[1, 2] 

Discussion

In line 198: “an U/S study” – please, introduce the abbreviation.

Addressed. Changed to ultrasound.

The statement in the sentence in lines 205-207 would need supporting literature reference or justification.

Addressed. Citation added.

There are no figure legends in the manuscript.

Addressed.

For approving a test for screening needs validation, first. Have to know what is the sensitivity, specificity, etc. of the new diagnostic method. And not only a comparison with healthy eyes but comparisons with other eye diseases are necessary to judge before the statement about the usefulness of the new diagnostic modality.

We mentioned that this tool might have potential screening and/or diagnostic value. To further clarify this claim, we qualified it with the statement that the tool needs another study dedicated for calculating sensitivity and specificity, with a larger sample size and with another research design in lines 291. Ultimately we aim make this method usable in the clinical setting, our team has started to collect a larger sample size and we are working on to improve and approve the test. If the statement is still objectionable we can omit it altogether.

We know that both retinal thickness and choroidal thickness depend on the axial eye length. The longer eyeballs have thinner layers and possibly with different reflectivity. What was the range of the axial eye length of the patients and controls? Were there differences in axial eye length between AE, FE and HE? Need to show if there is a correlation between axial eye length and the reflectivity of the layers.

To the best of our knowledge, axial length is only associated with a change in choroidal volume[3] and thickness[4] and not reflectivity. Axial length is associated with a change in RNFL reflectivity since RNFL reflectivity is directionally dependent[5] (myopes and hyperopes are very likely to have differences in cell bundle orientation). 

We tried to account for axial length by performing another set of analysis excluding myopes and strabismic participants. The results were very similar and consistent to previous statistical analysis that did not account for axial length, except for ROI#4 which included the choriocapillaris and the Sattler’s layer. This might be due to the smaller sample size and/or the difficulty in delineating the borders of ROI#4. It could also be due to changes in thickness and volume which could be speculatively affecting reflectivity. We added the statistical results in the results section and we mentioned this matter in the discussion. (lines 215-233,337-347) (all added parts including tables are highlighted in yellow in the revised manuscript)

1. Thiese MS, Ronna B, Ott U. P value interpretations and considerations. Journal of thoracic disease. 2016;8(9):E928-e31.

2. Sullivan GM, Feinn R. Using Effect Size-or Why the P Value Is Not Enough. Journal of graduate medical education. 2012;4(3):279-82.

3. Barteselli G, Chhablani J, El-Emam S, Wang H, Chuang J, Kozak I, et al. Choroidal volume variations with age, axial length, and sex in healthy subjects: a three-dimensional analysis. Ophthalmology. 2012;119(12):2572-8.

4. Flores-Moreno I, Lugo F, Duker JS, Ruiz-Moreno JM. The relationship between axial length and choroidal thickness in eyes with high myopia. American journal of ophthalmology. 2013;155(2):314-9.e1.

5. Huang XR, Knighton RW, Feuer WJ, Qiao J. Retinal nerve fiber layer reflectometry must consider directional reflectance. Biomed Opt Express. 2016;7(1):22-33.

---

## [Decision Letter · Decision Letter 1]

23 Jul 2021

Morphological changes in amblyopic eyes in choriocapillaris and Sattler’s layer in comparison to healthy eyes, and in retinal nerve fiber layer in comparison to fellow eyes through quantification of mean reflectivity: a pilot study.

PONE-D-21-10099R1

Dear Dr. Estephan,

We’re pleased to inform you that your manuscript has been judged scientifically suitable for publication and will be formally accepted for publication once it meets all outstanding technical requirements.

Kind regards,

Sanjoy Bhattacharya

Academic Editor

PLOS ONE

Additional Editor Comments (optional):

Reviewers' comments:

Reviewer's Responses to Questions

**Comments to the Author**

1. If the authors have adequately addressed your comments raised in a previous round of review and you feel that this manuscript is now acceptable for publication, you may indicate that here to bypass the “Comments to the Author” section, enter your conflict of interest statement in the “Confidential to Editor” section, and submit your "Accept" recommendation.

Reviewer #1: All comments have been addressed

2. Is the manuscript technically sound, and do the data support the conclusions?

Reviewer #1: Yes

3. Has the statistical analysis been performed appropriately and rigorously? 

Reviewer #1: Yes

4. Have the authors made all data underlying the findings in their manuscript fully available?

Reviewer #1: (No Response)

5. Is the manuscript presented in an intelligible fashion and written in standard English?

Reviewer #1: Yes

6. Review Comments to the Author

Reviewer #1: All comments were addressed and included in the manuscript. The present form of the manuscript is suitable for publication.

7. PLOS authors have the option to publish the peer review history of their article (what does this mean?). If published, this will include your full peer review and any attached files.

Reviewer #1: No

---

## [Editor Report · Acceptance letter]

28 Jul 2021

PONE-D-21-10099R1 

Morphological changes in amblyopic eyes in choriocapillaris and Sattler’s layer in comparison to healthy eyes, and in retinal nerve fiber layer in comparison to fellow eyes through quantification of mean reflectivity: a pilot study. 

Dear Dr. Estephan:

I'm pleased to inform you that your manuscript has been deemed suitable for publication in PLOS ONE. Congratulations! Your manuscript is now with our production department. 

Kind regards, 

on behalf of

Dr. Sanjoy Bhattacharya 

Academic Editor

PLOS ONE